# ^64^CuCl_2_ PET Imaging of 4T1-Related Allograft of Triple-Negative Breast Cancer in Mice

**DOI:** 10.3390/molecules27154869

**Published:** 2022-07-29

**Authors:** Adrien Latgé, Frédéric Boisson, Ali Ouadi, Gerlinde Averous, Lionel Thomas, Alessio Imperiale, David Brasse

**Affiliations:** 1Nuclear Medicine and Molecular Imaging Department, Institut de Cancérologie de Strasbourg Europe (ICANS), 17 Rue Albert Calmette, 67200 Strasbourg, France; a.imperiale@icans.eu; 2Institut Pluridisciplinaire Hubert Curien, Université de Strasbourg, 23 Rue du Loess, 67037 Strasbourg, France; frederic.boisson@iphc.cnrs.fr (F.B.); ali.ouadi@iphc.cnrs.fr (A.O.); lionel.thomas@iphc.cnrs.fr (L.T.); david.brasse@iphc.cnrs.fr (D.B.); 3CNRS, UMR7178, 23 Rue du Loess, 67037 Strasbourg, France; 4Department of Pathology, Hôpitaux Universitaires de Strasbourg, 1 Avenue Molière, 67200 Strasbourg, France; gerlinde.averous@chru-strasbourg.fr

**Keywords:** ^64^CuCl_2_, breast cancer, 4T1, biodistribution, PET/CT, ^18^F-FDG

## Abstract

^64^CuCl_2_ is an economic radiotracer for oncologic PET investigations. In the present study, we characterized the uptake of ^64^CuCl_2_ in vivo by µPET/CT in an allograft 4T1-related mouse model (BALB/c) of advanced breast cancer. ^18^F-FDG was used as a comparator. Twenty-two animals were imaged 7–9 days following 4T1-cell implantation inside mammary glands. Dynamic ^64^CuCl_2_ µPET/CT acquisition or iterative static images up to 8 h p.i. were performed. Animal biodistribution and tumor uptake were first evaluated in vivo by µPET analysis and then assessed on tissue specimens. Concerning ^18^F-FDG µPET, a static acquisition was performed at 15 min and 60 min p.i. Tumor ^64^CuCl_2_ accumulation increased from 5 min to 4 h p.i., reaching a maximum value of 5.0 ± 0.20 %ID/g. Liver, brain, and muscle ^64^CuCl_2_ accumulation was stable over time. The tumor-to-muscle ratio remained stable from 1 to 8 h p.i., ranging from 3.0 to 3.7. Ex vivo data were consistent with in vivo estimations. The ^18^F-FDG tumor accumulation was 8.82 ± 1.03 %ID/g, and the tumor-to-muscle ratio was 4.54 ± 1.11. ^64^CuCl_2_ PET/CT provides good characterization of the 4T1-related breast cancer model and allows for exploration of non-glycolytic cellular pathways potentially of interest for theragnostic strategies.

## 1. Introduction

Copper is a metallic element involved in the regulation of multiple physiological processes in humans, such as cell growth and neurotransmitter synthesis. After internalization by the high-affinity transmembrane protein hCTR-1, copper accumulates in intracellular organelles via various chaperone proteins, such as CCS, ATOX1, Cox17, SCO1, and SCO2 [1,2,3]. Copper seems to play a key role in tumor growth, neoangiogenesis, and metastatic spread [4,5,6]; overexpression of hCTR-1 in various cancers (prostate, lung, melanoma, glioblastoma, etc.) has been revealed by several studies [3].

Copper-64 (^64^Cu) is a cyclotron-produced radionuclide that is ideally suited for positron emission computed tomography (PET) investigation, enabling image acquisition over 48 h [7]. Copper radioisotopes such as ^64^Cu and ^67^Cu offer an interesting opportunity for diagnosis and related radiometabolic treatment. The use of ^64^Cu permits prospective dosimetry or dose personalization of ^67^Cu, opening new possibilities in the theragnostic field. ^64^Cu and ^67^Cu have optimal half-lives (12.7 h and 2.5 d, respectively) and energies appropriate for use with biologic drugs. Moreover, the energy of ^67^Cu is equivalent to that of ^177^Lu, which is routinely used for peptide receptor radionuclide therapy (PRRT) in patients with metastatic neuroendocrine tumors [8].

The ionic form of copper chloride (^64^CuCl_2_) offers economic and logistic advantages, with no need for complex chemistry to be produced, and can be directly administered in vivo in an aqueous solution. The results from clinical studies underline the feasibility of ^64^CuCl_2_ use as a metabolic radiotracer for PET/CT investigations in patients with prostate cancer relapse [9]. Several preclinical analyses have also been performed with promising results using cellular lines and animal models of prostate, colon, ovarian, lung, brain, and head-and-neck tumors. However, clinical and preclinical data concerning ^64^CuCl_2_ PET investigations in breast cancer are currently scarce. 4T1 is an interesting cell line derived from murine breast carcinoma [10] and can grow in culture or after allografting in immunocompetent mice. As a consequence, it is widely used as a surrogate for late-stage triple-negative breast cancer (TNBC) in humans due to its highly invasive potential and substantial molecular similarity (lack of Esr1, Erbb2, and Pgr expression) [11]. As a human TNBC model, 4T1-related tumors in mice have shown intense ^18^F-fluorodeoxyglucose (^18^F-FDG) uptake; thus, ^18^F-FDG PET is often used as a comparator in preclinical imaging studies [12]. Due to the recent development of therapeutics, several molecular alternatives investigating nonglycolytic pathways have been researched for breast cancer imaging [13,14]. To our knowledge, in no case has ^64^Cu been used as a labeling isotope or ^64^CuCl_2_ been used as a diagnostic agent for TNBC.

Therefore, in the present study, we have characterized the uptake of ^64^CuCl_2_ in an allograft 4T1-related mouse model of advanced breast cancer by in vivo PET. ^18^F-FDG PET was used as an image comparator.

## 2. Results

Twenty-nine female BALB/c mice were used for the study: 10 for ^64^CuCl_2_ PET investigations, 13 for ^64^CuCl_2_ ex vivo biodistribution assessment, and 6 for ^18^F-FDG PET images and ex vivo studies. Among the 22 mice imaged by µPET/CT, 6/22 underwent ^64^CuCl_2_ dynamic studies, 10/22 underwent ^64^CuCl_2_ static acquisitions, and 6/22 underwent ^18^F-FDG static imaging.

### 2.1. Tumor Size and Histology

Seven to nine days after 4T1 cells were injected into the mammary gland, tumors reached an average volume of 315 ± 175 mm^3^. Tumors developed locally without detection of extramammary metastases upon visual investigation. Pathological reports revealed hyperchromatic cells with polymorphic nuclei, restricted cytoplasms, and numerous mitoses. Tumors were positive for cytokeratin 5/6 (Figure 1).

### 2.2. ^64^CuCl_2_ Dynamic µPET/CT Acquisitions

Six mice were imaged from 5 to 60 min after intravenous (i.v., *n* = 3) or intraperitoneal (i.p., *n* = 3) ^64^CuCl_2_ administration (Figure 2). Activity inside tumors and organs was higher for intravenous injection. The tumor SUVmean increased linearly from 0.75 ± 0.16 at 5 min p.i. to 1.12 ± 0.23 at 60 min p.i. for i.v. injected mice; the corresponding percentages of injected doses per gram were 3.50 ± 0.37 and 5.20 ± 0.31, respectively. The same trend was observed in mice that underwent i.p. injections (0.79 ± 0.57 to 2.68 ± 0.63 %ID/g at 5 and 60 min p.i., respectively). The tumor mean SUV was higher in animals that underwent intravenous ^64^CuCl_2_ injection (5 min p.i, *p* < 0.01). The liver and brain displayed a slight activity increase within the first hour after injection: SUVmean varied from 4.52 ± 0.32 (liver) and 0.47 ± 0.01 (brain) at 5 min p.i. to 4.95 ± 0.22 (liver) and 0.77 ± 0.34 (brain) at 60 min p.i. for i.v. injected mice. In contrast, muscle showed a globally stable SUVmean over time (from 0.37 ± 0.12 at 5 min p.i. to 0.34 ± 0.08 at 60 min p.i.) for i.v. injected mice, while there was increasing activity over time for i.p. injected mice (0.03 ± 0.01 at 5 min p.i. to 0.18 ± 0.02 at 60 min p.i.). Even in these cases, SUV values were higher in animals that underwent i.v. injection (5 min p.i, *p* < 0.04). Kidney biodistribution was somewhat different: there was a significant decrease in the mean SUV from 5 to 60 min p.i. (*p* = 0.03) in mice with i.v. injection (corresponding to 15.6 ± 2.1 to 9.1 ± 0.6 %ID/g at 5 min and 60 min p.i., respectively), while activity remained stable in mice with i.p. injection.

### 2.3. ^64^CuCl_2_ µPET/CT Static Acquisitions

Four i.v. injected mice were imaged from 1 h to 8 h p.i. (Figure 3). A mild but significant increase in activity was observed within tumors. SUVmean ranged from 0.81 ± 0.12 at 1 h p.i. to a maximum of 1.02 ± 0.08 at 4 h p.i. (*p* = 0.02), corresponding to 5.0 ± 0.20 %ID/g. The liver had the highest ^64^CuCl_2_ accumulation, showing an SUVmean from 3.78 ± 1.03 at 1 h p.i. to 4.35 ± 0.43 at 4 h p.i. (18.5 ± 5.6 to 21.3 ± 2.8 %ID/g). Brain and muscle activity variation was minimal from 1 h to 8 h p.i., showing the highest activity at 8 h p.i. of 1.5 ± 0.1 %ID/g for brain and 1.6 ± 0.26 %ID/g for muscle. The tumor-to-muscle ratio exhibited a nonsignificant increase from 1 h (3.13 ± 0.32) to 4 h p.i. (3.71 ± 0.64) (*p* = 0.23).

### 2.4. ^64^CuCl_2_ Ex Vivo Biodistribution Study

Ex vivo biodistribution studies were conducted in 13 mice with tumor grafts and sacrificed at 1 h (*n* = 5), 4 h (*n* = 4), and 8 h (*n* = 4) after ^64^CuCl_2_ i.v. administration (Figure 4). The mean weight of the examined tissue specimens was 0.36 ± 0.03 g for the brain, 0.11 ± 0.02 g for the heart, 0.19 ± 0.05 g for the lungs, 0.24 ± 0.07 g for the liver, 1.95 ± 0.55 g for the intestine, 0.16 ± 0.02 g for the kidney, and 0.13 ± 0.05 g for the muscle. The mean excised tumor weight was 0.19 ± 0.06 g.

No significant variation in tumor activity was observed from 1 h p.i. (4.52 ± 3.07 %ID/g) to 8 h p.i. (4.47 ± 1.42 %ID/g) (*p* = 0.98). The liver had the highest ^64^CuCl_2_ accumulation (20.10 ± 5.77 %ID/g at 4 h p.i.). The intestine was the only organ displaying an important temporal variation in ^64^CuCl_2_ uptake mainly due to intestinal excretion (3.10 ± 1.88 %ID/g at 1 h, 21.16 ± 4.92 %ID/g at 8 h p.i.). Kidney cortical activity averaged 10.15 ± 1.91 %ID/g at 1 h p.i (right kidney measurement), with a slight decrease over time without reaching significance (*p* = 0.22). The brain and muscle were characterized by the lowest tracer accumulation: 0.31 ± 0.18 %ID/g and 0.48 ± 0.21 %ID/g (1 h p.i.), respectively.

### 2.5. ^18^F-FDG µPET/CT and Ex Vivo Biodistribution Study

Six mice were scanned at 15 and 60 min after ^18^F-FDG i.v. administration and then sacrificed and dissected for the biodistribution study. The tumor-to-muscle ^18^F-FDG uptake ratios were 2.11 ± 0.31 at 20 min and 4.54 ± 1.11 at 65 min p.i. Tumor SUVmean values were 0.69 ± 0.07 at 20 min p.i. and 1.94 ± 0.25 at 65 min p.i. SUVmean at 65 min p.i. of brain and muscle were 1.11 ± 0.11 and 0.44 ± 0.08, respectively. From ex vivo measurements, the mean tumor ^18^F-FDG accumulation at 70 min p.i. was 8.82 ± 1.03 %ID/g.

## 3. Discussion

Our study first assessed the uptake profile of ^64^CuCl_2_ in a 4T1 allograft mouse model of TNBC considering ^18^F-FDG as a comparator. ^64^CuCl_2_ accumulation was more than 3-fold higher than that in the muscle background, allowing clear tumor detection and characterization in all examined animals. ^64^CuCl_2_ tumor accumulation was maximal approximately 4 h after ^64^CuCl_2_ administration, and tumors were detectable from 5 min to 8 h p.i. The tumor accumulation of ^64^CuCl_2_ remains stable and close to 5 %ID/g at 1 h, 4 h, and 8 h p.i. These results are encouraging and support the feasibility of ^64^CuCl_2_ as a PET radiotracer for 4T1-related TNBC imaging.

Copper-64 is most often used to label biologically active molecules such as proteins, antibodies, and nanoparticles through bifunctional chelators (DOTA, TETA). Nonetheless, this approach is complex, requires highly specific activities, and has numerous technical drawbacks, such as poor in vivo stability and chemical reactions with serum proteins. Conversely, copper-64—in its ionic form—achieves high in vivo stability, does not require complex radiochemistry, is not involved in degradation issues, and can be easily available through cyclotron-based production [15].

The 4T1 tumor cell line was previously screened as a potential candidate among 12 different tumor lines in a study designed to evaluate the diagnostic value of ^64^CuCl_2_ [16]. Assuming that ^64^CuCl_2_ uptake would be correlated with the quantity of the main copper transporter, each of these cell lines was tested in vitro for CTR-1 expression. The CTR-1/actin ratio ranged from approximately 0.1 (PC-3) to >1.0 (MDA-MB-435). The expression ratio of the 4T1 cell line was approximately 0.5, suggesting moderate overexpression of CTR-1 that, according to our observations, seems to be sufficient for tumor visualization by ^64^CuCl_2_ PET studies.

Jørgensen et al. evaluated ^64^CuCl_2_ binding among 5 human tumor lines (A2780, FaDu, H727, HT-29, U87MG) xenografted into NMRI mice. In vivo imaging was performed at 1 h and 22 h p.i., and the results were correlated with CTR-1 expression [17]. At 1 h p.i., ^64^CuCl_2_ activity ranged from 1.53 ± 0.07 %ID/g (A2780) to 3.48 ± 0.35 %ID/g (U87MG). FaDu, H727, and U87MG cell lines showed a significant ^64^CuCl_2_ uptake increase over time (up to 4.92 ± 0.17 for H727), but no correlation was found between ^64^CuCl_2_, tumor size, and CTR-1 expression. In our study, ^64^CuCl_2_ biodistribution remained stable between 1 h and 8 h p.i., allowing PET imaging to be performed during a large timeframe. Moreover, the important ^64^CuCl_2_ accumulation in 4T1 allografts (4.52 ± 3.07 %ID/g as early as 1 h p.i) suggests that TNBC is among the tumors with the highest ^64^CuCl_2_ avidity.

As ^18^F-FDG is the preferred radiotracer for TNBC [18], it was important to assess to what extent ^64^CuCl_2_ could provide satisfying PET images compared to ^18^F-FDG in 4T1-bearing mice. All tumors were clearly visualized by ^64^CuCl_2_ and ^18^F-FDG PET (Figure 5). In both cases, a photopenic area was present at the center of the largest tumors. This pattern is well known in PET clinical practice and is usually attributed to altered vascularization and tissue necrosis inside voluminous tumoral lesions.

Despite good tumor visualization, ^18^F-FDG accumulation was almost 2-fold higher than ^64^CuCl_2_ uptake (8.82 %ID/g at 65 min p.i. vs. 4.70 %ID/g at 4 h p.i.). However, the interest of ^64^CuCl_2_ could potentially reside in its ability to act as a therapeutic agent due to its β^−^ and Auger electron emission (in addition to diagnostic β^+^ emission). This hypothesis is based on data suggesting that nuclear Cu^+^ internalization is mediated by ATOX1 only in p53-positive cell lines [19]. This behavior allows for the potential therapeutic application of radioactive copper [20,21]. Ferrari et al. investigated a glioblastoma–astrocytoma animal model in athymic nude male BALB/c mice by ^64^CuCl_2_ PET [20]. Adequate tumor PET visualization was achieved after the injection of 12 MBq of ^64^CuCl_2,_ reaching a tumor-to-healthy tissue ratio of 2.88, similar to what we have observed with 4T1-related tumors. After therapeutic injection of ^64^CuCl_2_ (single 333 MBq injection or 6 daily injections of 55.5 MBq), the tumor volume showed a reduction of 64–94% between the 1st and 20th weeks after injection. The survival rate at the 20th week drastically improved, from 0% in untreated mice to 73% in mice treated with multiple injections of ^64^CuCl_2_. Hematological toxicity (40% reduction in leukocytes) was moderate and reversible 5 weeks after ^64^CuCl_2_ administration.

The biodistribution of copper in humans has been widely studied. Ingested copper is absorbed by the proximal intestine, travels through the portal vein and undergoes a first hepatic passage. Once inside the blood, the copper circulates in the plasma compartment and is mostly bound to ceruloplasmin (60–95%) and albumin (10%). Its elimination is mainly hepato-biliary, explaining the diffuse hyperfixation of the liver and intestines [22]. As previously reported, kidneys displayed an important accumulation of ^64^CuCl_2_. This fact could be linked to the presence of superoxide dismutase, a ubiquitous enzyme present in the cytosol of eukaryotic cells and abundant in the liver and kidney, for which copper acts as a cofactor [23]. Renal uptake was predominant in the cortex, without significant bladder filling over time, rejecting the hypothesis of ^64^CuCl_2_ urinary elimination [24]. Relative ^64^CuCl_2_ kidney uptake was even higher from 30 min to 3 h p.i. in the study from Manrique-Arias et al. in Wistar rats [25]. These data contrast with our observations, as kidneys were only the third most accumulating organ after the liver and intestines. Using ^64^CuCl_2_ for therapeutic purposes would lead to higher injected doses, and identifying dose-limiting organs would be of paramount importance, as radionuclide therapy may lead to a wide spectrum of radiation-induced side effects [26].

Radiomic analysis on PET tumoral data could improve biological characterization using artificial intelligence frameworks. With the help of machine-learning, non-FDG PET explorations may lead to better understanding of tumoral features and related prognosis factors [27]. In vivo theranostic strategies could also benefit from nanomedicines, able to carry both diagnostic (e.g., ^64^Cu for PET, Gd(III) for MRI) and therapeutic compounds [28].

The main limitation of our work is the lack of complementary immunohistochemical characterization of 4T1 tumors. Determining the expression level of several proteins involved in copper uptake, transport, and efflux (particularly CTR-1, ATP7A/B, ATOX1) remains necessary to define the mechanism of ^64^CuCl_2_ uptake. A better understanding of the underlying molecular mechanisms should help to assess which pathways are involved in ^64^Cu homeostasis. To evaluate the potential role of ^64^CuCl_2_ for therapeutic purposes, ATOX-mediated Cu^+^ internalization within the nucleus should be further investigated.

## 4. Materials and Methods

### 4.1. Radiotracer Production

For ^64^CuCl_2_ production, the target material isotopically enriched ^64^Ni was purchased from ChemGas (Boulogne, France). Ni plating was performed based on the method reported by Wesley and Carey [29]. Target irradiation was performed using a TR24 cyclotron from ACSI, which delivered an extracted energy proton beam between 16 MeV and 25 MeV. A dedicated degrader was developed to obtain proton energy in the range of 11 MeV on target (optimal production yield of ^64^Cu for thin target). A few hours after bombardment, the plated Ni layer was dissolved in a concentrated 9 M HCl solution. The solution was transferred onto a 1.5 × 15 cm AG1-X8 anion exchange column. The column was washed with 6 M HCl solution to collect ^64^Ni. Then, 0.1 M HCl was added to elute ^64^Cu in ^64^CuCl_2_ form. Using a heating block, the eluted ^64^CuCl_2_ was evaporated to dryness and then diluted in saline solution (pH > 5.5). The good-manufacturing-practices-compliant ^18^F-FDG was provided by Curium (Dijon, France).

### 4.2. 4T1 Cell Culture and Animal Model

4T1 cells were obtained from Weizmann Institute of Science (Rehovot, Israel). Culture was performed under the following conditions: constant temperature (37 °C), 5% CO_2_, DMEM with 4.5 g of glucose supplemented with 10% fetal bovine serum and 1% antibiotics (penicillin/streptomycin). A suspension of 10^6^ 4T1 cells in 100 µL of PBS was injected into the mammary gland of 29 female BALB/c mice after anesthesia inhalation (2% isoflurane). Tumor width and length were measured with a caliper to calculate the tumor volume. Animals were maintained at constant temperature (22 °C) and humidity (40%) with 12 h light/dark cycles and unlimited access to water and food. Individual cages were ventilated with HEPA filters in the dedicated IPHC facility in Strasbourg.

### 4.3. PET/CT Imaging Acquisition and Analysis

PET imaging was performed 7 to 9 days after 4T1 cell allografting. All PET/CT acquisitions were performed on an IRIS PET/CT device (Inviscan, Strasbourg, France) equipped with 2 octagonal rings of 8 block detectors each, a matrix of 702 LYSO:Ce crystals of 1.6 × 1.6 × 12 mm each per module, and 64 Hamamatsu H8500 photomultiplier anodes. The geometrical configuration provides an axial coverage of 95 mm and a transverse field of view of 80 mm. The spatial resolution of the system is below 1.5 mm, with an absolute sensitivity close to 8% (NEMA standard) [30].

The ^64^CuCl_2_ PET acquisition protocol included a dynamic phase of 11 frames of 5 min each starting 5 min after the injection of 351 ± 56 kBq/g (^18^F equivalent) of ^64^CuCl_2_ and a 10-min static image acquired at 1 h, 2 h, 4 h, and 8 h p.i. ^64^CuCl_2_ was injected into the mouse tail vein (intravenous) or inside the peritoneum to compare both injection modalities. Concerning ^18^F-FDG PET, a 10-min static acquisition was performed at 15 min and 60 min after the injection of 278 ± 45 kBq/g radiotracer in the tail vein. All ^64^CuCl_2_ and ^18^F-FDG PET data were reconstructed using the OSEM algorithm (8 iterations, 8 subsets) in a 3D volume of 201 × 201 × 120. CT acquisition (80 kV, 0.9 mA) was performed for anatomical correlation. PET data were corrected for random coincidences. No attenuation or scatter corrections were applied. The results of PET images were qualitatively interpreted as positive or negative on a dedicated workstation (Syngo.via VB30; Siemens, Erlangen, Germany). A focal nonphysiologic increase in radiotracer uptake in the area of tumor development was considered to be a positive PET result. For a semiquantitative assessment of tumoral uptake, an elliptic volume of interest (VOI) was drawn to assess the mean activity of each organ and tumor. Muscle activity was determined from the left posterior triceps. Finally, time–activity curves were extracted from VOIs for the entire dynamic PET acquisition. SUVs were calculated from the mean voxel value within the VOIs. SUVs were recorded and used to generate time–activity curves.

To obtain ^64^CuCl_2_ biodistribution over time, 13 4T1-bearing BALB/c mice were sacrificed and dissected at 1 h (*n* = 4), 4 h (*n* = 5), or 8 h (*n* = 4) postinjection (p.i.). After excision of the tumor and organs of interest (liver, brain, intestines, kidneys, lungs, heart, and muscle), each histological specimen was weighed and placed in an automatic gamma counter (Hidex, Turku, Finland) for 1 min. The count obtained was converted to absolute activity using a calibration curve and then corrected for radioactive decay to an activity value arbitrarily taken at the time of animal sacrifice. The results are expressed as a percentage of the injected dose per gram of tissue (%ID/g).

### 4.4. Data Presentation

The quantitative data are reported as the mean values and standard deviations. Box plots were used for graphical representation of the results. Means were compared using Student’s *t*-test (for paired and independent values). *p* < 0.05 was considered significant. Statistical analysis was performed using the online BiostaTGV (https://biostatgv.sentiweb.fr, accessed on 10 March 2022).

## 5. Conclusions

^64^CuCl_2_ is easily available through cyclotron-based production and can be efficiently used as a PET radiotracer in 4T1-related TNBC. Although ^64^CuCl_2_ uptake is lower than ^18^F-FDG uptake, the method allows for nonglycolytic molecular characterization of TNBC in an animal model. Complementary studies are needed to correlate ^64^CuCl_2_ uptake to cell expression of proteins involved in copper transportation, particularly to enable theragnostic strategies.

## Figures and Tables

**Figure 1 molecules-27-04869-f001:**
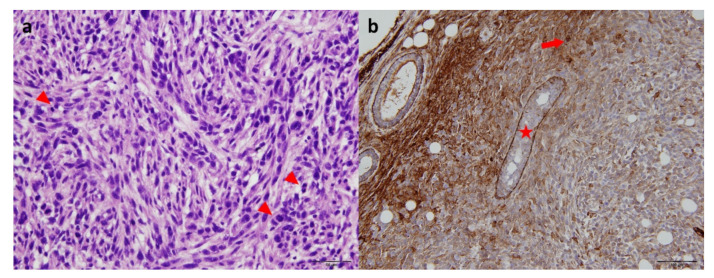
Histological sections of 4T1-related allografts. (**a**) Hematoxylin-eosin staining (×400) showing fusocellular patterns with numerous mitoses (arrowheads). (**b**) Cytokeratin 5/6 staining (×200) revealing tumoral cells (arrow) and basal cells (star).

**Figure 2 molecules-27-04869-f002:**
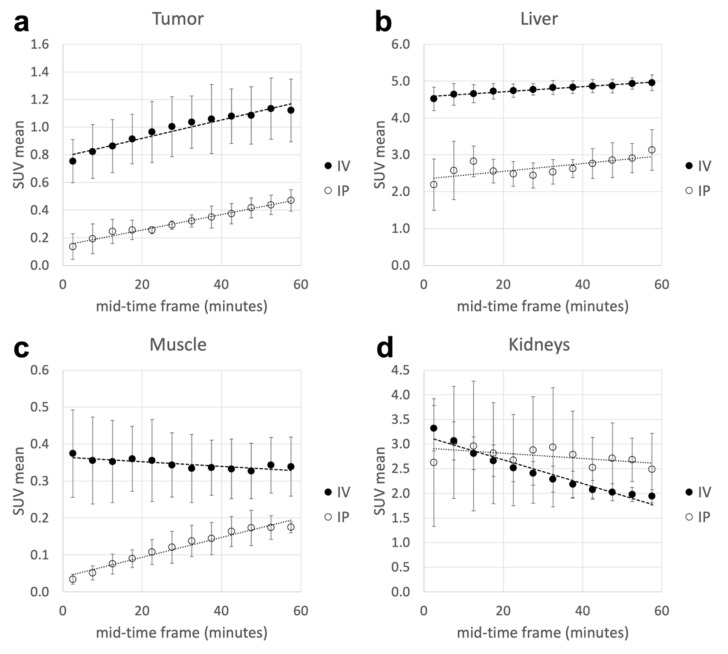
In vivo dynamic PET assessment (from 5 to 60 min p.i.) of ^64^CuCl_2_ activity expressed as SUVmean inside mammary tumor and organs of interest. (**a**) Tumor. (**b**) Liver. (**c**) Muscle. (**d**) Kidneys. IV: intravenous; IP: intraperitoneal.

**Figure 3 molecules-27-04869-f003:**
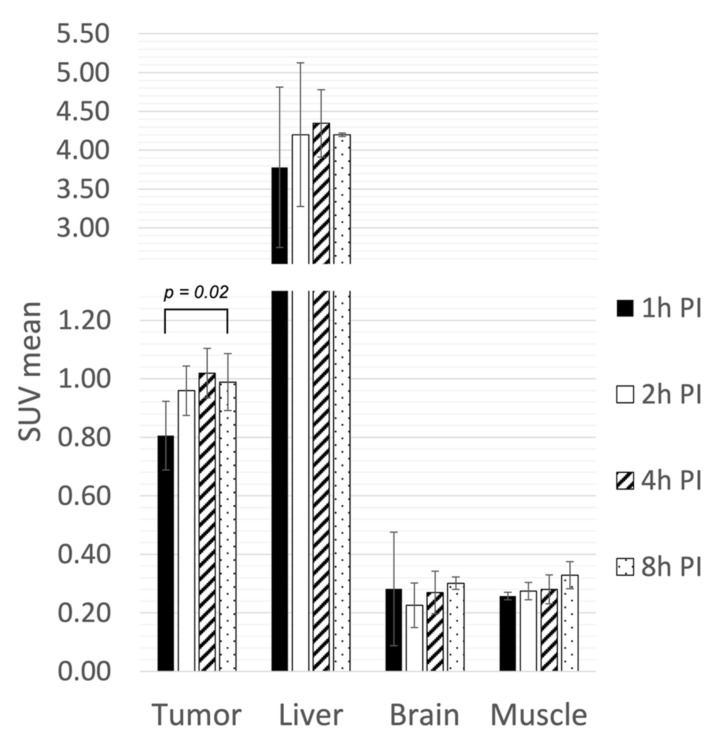
In vivo static PET assessment (from 1 to 8 h p.i.) of ^64^CuCl_2_ activity expressed as SUVmean inside mammary tumor and organs of interest. IV: intravenous; IP: intraperitoneal.

**Figure 4 molecules-27-04869-f004:**
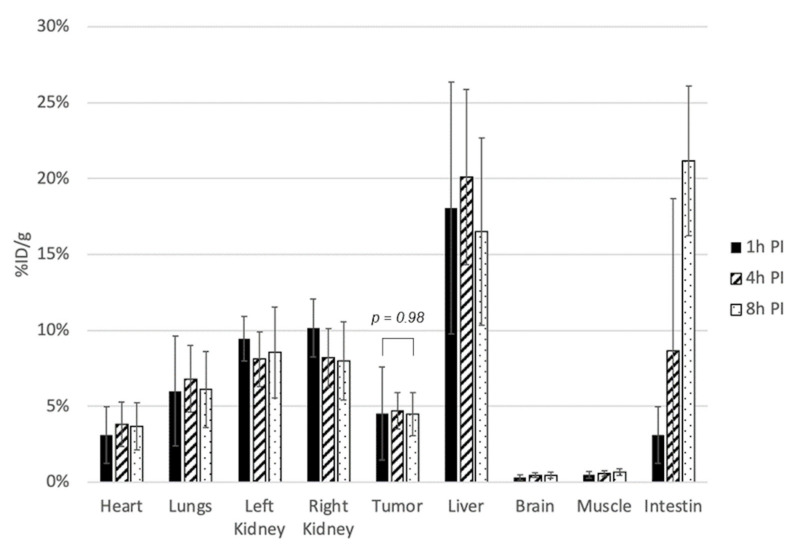
Ex vivo assessment of ^64^CuCl_2_ activity in tumors and organs of interest at 1 h, 4 h, and 8 h p.i.

**Figure 5 molecules-27-04869-f005:**
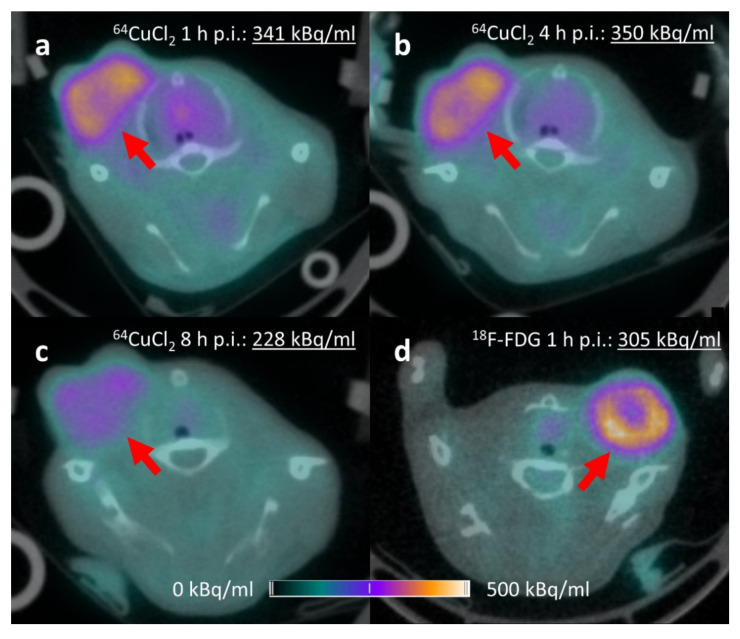
Mammary tumors assessed with ^64^CuCl_2_ PET/CT over time, with ^18^F-FDG PET/CT as a reference. ^64^CuCl_2_ tumor uptake expressed as mean kBq/mL inside VOI was higher at 4 h p.i. and slightly reduced over time. (**a**) Axial fused ^64^CuCl_2_ PET/CT, 1 h p.i. (**b**) Axial fused ^64^CuCl_2_ PET/CT, 4 h p.i. (**c**) Axial fused ^64^CuCl_2_ PET/CT, 8 h p.i. (**d**) Axial fused ^18^F-FDG PET/CT, 1 h p.i.

## Data Availability

Not applicable.

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
