# Peer review of "^64^CuCl_2_ PET Imaging of 4T1-Related Allograft of Triple-Negative Breast Cancer in Mice"

_molecules, 2022, doi:10.3390/molecules27154869_

Round 1

Reviewer 1 Report

1. Quality of Figure 2 is too low to identify useful information as a reviewer;

2. Why not fitting the data in Figure 2? Maybe this mathematic operation could provide more useful information to well understand dynamics of this reagents in vivo

3. Why not provide statical information in relative figures for example Figure 2d, Figure 3.

4.  What about the blood stability of these reagent and whether to bind with serum can shelter the signal of reagent in vivo or in vitro? maybe author should provide in vitro assay to examine this hypothesis/

5. Why not provide the quantitatively measurement of in vivo imaging results? for example in Figure 4. Time-dependent manner or time course response may provide more useful information to support your conculsion

6. Pay more attention about the formation of presented data, for example blank should be existed between Number and unit, and “1.5×1.5” should be instead of “1.5x1.5” (not x, should be “Multiplication sign”)

7. citations in this paper is not enough to support this conclusion, please cite more references, for examples as follows:

a. Physics in Medicine & Biology, 2014, 59(18): R233.

b. https://doi.org/10.1002/EXP.20210134

c. https://doi.org/10.1002/EXP.20210222

d. Iyer R, Jhingran A. Radiation injury: imaging findings in the chest, abdomen and pelvis after therapeutic radiation[J]. Cancer Imaging, 2006, 6(Spec No A): S131.

e. Thompson R F, Valdes G, Fuller C D, et al. Artificial intelligence in radiation oncology imaging[J]. International Journal of Radiation Oncology, Biology, Physics, 2018, 102(4): 1159-1161.

Reviewer 2 Report

The manuscript studied the in vivo uptake of 64CuCl2 in mouse model of advanced breast cancer by PET and concluded that 64CuCl2 PET/CT provides good characterization of the 4T1-related breast cancer 29model and allows for exploration of non-glycolytic cellular pathways potentially of interest for 30theragnostic strategies.

Questions or concerns:

1. Line 67: “To our knowledge, in no case has 64Cu been used as a labeling isotope or 64CuCl2 been used as a diagnostic agent.” There are some papers about the use of 64CuCl2 in prostate cancer diagnostics and staging in clinical trials. (such as PMIDs: 25833290, 35464672)

2. The quality of Figure 2 is low, not clear enough.

Round 2

Reviewer 1 Report

Authors have well answered my concern. I recommended to publish